A comparison of deep transfer learning backbone architecture techniques for printed text detection of different font styles from unstructured documents

Mahadevkar Supriya 1
Patil Shruti shruti.patil@sitpune.edu.in 2
Kotecha Ketan 2
Abraham Ajith 3 4
1 Symbiosis Institute of Technology, Symbiosis International (Deemed University) , Pune , India
2 Symbiosis Centre for Applied Artificial Intelligence, Symbiosis Institute of Technology Symbiosis International (Deemed University) , Pune , Maharashtra , India
3 School of Computer Science Engineering & Technology, Bennett University , Greater Noida , Uttar Pradesh , India
4 Innopolis University , Innopolis , Republic of Tatarstan , Russia
Shanmuganathan Vimal
Electronic publication date: 2024 Feb 23
Publication date: 2024
Volume: 10
Electronic Location ID: e1769
Received 2023 Aug 3; Accepted 2023 Nov 30
Copyright: ©2024 Mahadevkar et al.
Copyright year: 2024
Copyright holder: Mahadevkar et al.
License: This is an open access article distributed under the terms of the Creative Commons Attribution License, which permits unrestricted use, distribution, reproduction and adaptation in any medium and for any purpose provided that it is properly attributed. For attribution, the original author(s), title, publication source (PeerJ Computer Science) and either DOI or URL of the article must be cited.
License URL: https://creativecommons.org/licenses/by/4.0/

Keywords: Printed text recognition, Object detection, Object recognition, Transfer learning, Backbone architectures, OCR (Object Character Recognition), ResNet50V2, Inception, Xception, VGG19

Funding: The Analytical Center for the Government of Russian Federation 70-2021-00143 IGK 000000D730321P5Q0002 This work was supported by the Analytical Center for the Government of Russian Federation, in 1 November 2021, under Grant 70-2021-00143 and Grant IGK 000000D730321P5Q0002. The funders had no role in study design, data collection and analysis, decision to publish, or preparation of the manuscript.

==============================
Object detection methods based on deep learning have been used in a variety of sectors including banking, healthcare, e-governance, and academia. In recent years, there has been a lot of attention paid to research endeavors made towards text detection and recognition from different scenesor images of unstructured document processing. The article’s novelty lies in the detailed discussion and implementation of the various transfer learning-based different backbone architectures for printed text recognition. In this research article, the authors compared the ResNet50, ResNet50V2, ResNet152V2, Inception, Xception, and VGG19 backbone architectures with preprocessing techniques as data resizing, normalization, and noise removal on a standard OCR Kaggle dataset. Further, the top three backbone architectures selected based on the accuracy achieved and then hyper parameter tunning has been performed to achieve more accurate results. Xception performed well compared with the ResNet, Inception, VGG19, MobileNet architectures by achieving high evaluation scores with accuracy (98.90%) and min loss (0.19). As per existing research in this domain, until now, transfer learning-based backbone architectures that have been used on printed or handwritten data recognition are not well represented in literature. We split the total dataset into 80 percent for training and 20 percent for testing purpose and then into different backbone architecture models with the same number of epochs, and found that the Xception architecture achieved higher accuracy than the others. In addition, the ResNet50V2 model gave us higher accuracy (96.92%) than the ResNet152V2 model (96.34%).

Introduction

Printed text recognition enables the conversion of physical documents into digital formats. Almost 90 to 95 percent of data generated on daily basis is in the form of unstructured data (Baviskar et al., 2021). Therefore, detecting and recognizing printed data from unstructured documents become important research area. This process allows for easy storage, retrieval, and sharing of information. By digitizing documents, organizations can eliminate the need for physical storage space, reduce paper waste, and facilitate efficient information management. AI, machine learning, and deep learning techniques have revolutionized printed text recognition by enabling more accurate and efficient recognition models (Mahadevkar et al., 2022). These techniques can be trained on large datasets, utilizing deep neural networks to handle complex patterns and variations in printed text.

The state-of-the-art of printed text recognition has made significant progress, but still faces several challenges. Some challenges include (Baviskar et al., 2021):

• Variability in fonts and languages: different fonts and languages exhibit varying styles and structures, making it challenging to create a universal recognition system that works accurately across diverse text types.

• Complex layouts: printed text appears in various layouts, such as newspapers, magazines, or books. Recognizing text accurately within complex layouts, with varying font sizes, columns, and orientations, is a persistent challenge.

• Noise and distortions: printed text may be affected by noise, distortions, or artifacts during the printing process. This can include smudges, ink bleed, or uneven printing, making it difficult for recognition systems to accurately interpret the text.

• Handwriting recognition: distinguishing between printed text and handwriting poses a challenge. Integrating robust handwriting recognition with printed text recognition in the same system is a complex task.

• Low-resolution images: recognizing text from low-resolution images or scanned documents is challenging, as it may result in loss of details and affect the accuracy of optical character recognition (OCR) systems.

• Ambiguous characters: some characters may look similar, leading to ambiguities in recognition. Resolving these ambiguities accurately is crucial for the reliability of text recognition systems.

• Document skew and distortion: scanned documents may exhibit skew or distortion, requiring preprocessing steps to rectify these issues before accurate text recognition can take place.

• Real-time processing: some applications, such as augmented reality or mobile scanning, require real-time text recognition. Achieving high accuracy in real-time scenarios with limited computational resources can be challenging.

• Handling special characters and symbols: recognition of special characters, mathematical symbols, or non-alphanumeric characters poses additional challenges, especially in scientific or technical documents.

• Adaptability to new fonts and languages: training OCR models to adapt to new fonts or languages requires substantial labeled data and can be time-consuming. Systems that can easily adapt to new linguistic and typographic variations are desired.

• Lack of standardized evaluation metrics: the absence of standardized metrics for evaluating text recognition systems can make it challenging to compare the performance of different models objectively.

With advancements in algorithms and computing power, they have proven to be highly effective in addressing the challenges associated with printed text recognition from unstructured documents. With pre-train models’ backbone architectures can also be very helpful to achieve more accurate results in case of printed text recognition from unstructured document processing. Transfer learning-based backbone architectures offer several advantages, including leveraging pre-trained models, generalization to unseen data, powerful feature extraction capabilities, end-to-end learning, superior performance, and adaptability, making them a preferred choice for printed text recognition from forms or invoices. Figure 1 illustrates the stepwise pipeline followed for printed text recognition from unstructured document processing (Jaiswal, 2021). The Backbone feature extractor networks capture features from the input image, and the easy decoder module of DeepLab models upsamples these features for creating segmented masks. Text object detection from document images is a computer vision task that involves identifying and localizing text objects of interest within an image (Rahman et al., 2020). Various backbone architectures have been used for object detection, each offering different strengths in terms of accuracy, speed, and efficiency (Patil et al., 2022).

Figure 1 Pipeline used for printed text processing from unstructured document processing.

Choosing a backbone architecture for printed text recognition involves considering factors such as model complexity, computational efficiency, and the specific characteristics of the dataset. Each of the mentioned architectures—ResNet50V2, Inception, Xception, and VGG19—comes with its own set of advantages, and the choice may depend on the nature of the printed text dataset and the requirements of the recognition task. Following are some popular backbone architectures used in printed text recognition (Zhang et al., 2020):

1. Residual neural network (ResNet): resNet is a widely used backbone architecture that introduced the concept of residual learning. It addresses the problem of vanishing gradients via skip connections, making it possible to train incredibly deep neural networks. Object detection frequently uses ResNet variations like ResNet-50, ResNet-101, and ResNet-152. Faster R-CNN with ResNet-101: region-based convolutional neural network, or faster R-CNN, is an appreciated object identification system. It uses a classifier to categorize and improve the ideas after using a region proposal network (RPN) to create prospective object areas. The Faster R-CNN framework can be combined with the ResNet-101 backbone for accurate and efficient object detection.

Advantages:

• Residual connections help mitigate the vanishing gradient problem, allowing for the training of deeper networks.

• ResNet architectures are known for their excellent performance in various computer vision tasks. ResNet50V2 specifically is a good balance between depth and computational efficiency.

• Residual connections can be beneficial in capturing long-range dependencies, which might be useful in understanding the context of text.

2. Visual Geometry Group (VGG): VGG is another widely adopted backbone architecture known for its simplicity and effectiveness. It is made up of max-pooling layers that are followed by a sequence of stacked convolutional layers with limited receptive fields. VGG variants, such as VGG16 and VGG19, have been used as backbone networks in object detection.

Single Shot MultiBox Detector (SSD) with VGG16: SSD is a real-time object detection method that performs object localization and classification in a single pass. By using a series of convolutional layers, SSD can generate a set of fixed-size bounding boxes at multiple scales. The VGG16 architecture has been employed as the backbone network in SSD, providing good detection accuracy, SSD can generate a set of fixed-size bounding boxes at multiple scales. The VGG16 architecture has been employed as the backbone network in SSD, providing good detection accuracy and real-time performance.

Advantages:

• VGG architectures are known for their simplicity and ease of understanding.

• The uniform structure with small convolutional filters can capture fine details.

• VGG19′s simplicity and uniformity might be advantageous for recognizing printed text with consistent font sizes and styles.

3. Inception (GoogLeNet): Inception, also known as GoogLeNet, introduced the concept of inception modules that utilize parallel convolutional operations of different filter sizes. These modules capture features at multiple scales and have a more efficient parameterization compared to larger convolutional kernels. GoogLeNet variants, such as InceptionV3 and InceptionResNetV2, have been utilized for object detection.

Region-based Fully Convolutional Network (R-FCN) with InceptionV2: R-FCN is an object detection framework that achieves high accuracy with shared computation across regions. It uses position-sensitive score maps to obtain class scores for each region proposal. In the case of R-FCN, InceptionV2 has been employed as the backbone network, offering a good balance between accuracy and efficiency.

Advantages:

• Inception architectures use multiple parallel paths of convolutions of different sizes, allowing the network to capture features at various scales.

• It balances computational efficiency and model performance.

• The multiple parallel convolutions in Inception can help in capturing both fine and coarse details in printed text, which may have variations in font size and style.

4. EfficientNet: A series of convolutional neural network topologies called EfficientNet enables more effective and state-of-the-art performance. The depth, width, and resolution of the network are scaled equally by these architectures using a compound scaling technique. EfficientNet models, such as EfficientNet-B0 to EfficientNet-B7, have gained popularity in various computer vision tasks, including object detection (Elharrouss et al., 2022). EfficientDet with EfficientNet-B4: EfficientDet is an efficient and accurate object detection framework that combines EfficientNet as the backbone network with a feature pyramid network (FPN) and a modified BiFPN (Bi-directional Feature Pyramid Network). EfficientDet models, such as EfficientDet-D0 to EfficientDet-D7, offer a wide range of trade-offs between speed and accuracy. EfficientNet-B4 is often chosen as the backbone for a good balance between performance and computational cost.

5. MobileNet: A lightweight framework called MobileNet was created for mobile and embedded devices. It utilizes depthwise separable convolutions along with pointwise convolutions to reduce model size and computational complexity while maintaining reasonable accuracy. Application in object detection: MobileNet has been used as a backbone architecture in object detection models, particularly in single-shot detectors like SSD and EfficientDet. Its lightweight nature enables fast inference on resource-constrained devices.

6. Xception: Xception is an addition to the Inception architecture that uses depthwise separable convolutions in place of the normal convolutional layers. The spatial and channel-wise convolutions are divided into independent operations by depth-wise separable convolutions, which speeds up computation and improves model performance. Application in object detection: Xception has been utilized as a backbone in object detection models, such as the EfficientDet framework. Its efficient design makes it suitable for real-time object detection tasks where computational resources are limited.

Advantages:

• Xception is an extension of Inception and replaces the standard convolutional layers with depth-wise separable convolutions, leading to increased representational efficiency.

• The depth-wise separable convolutions in Xception can be beneficial for capturing intricate details in printed text, which may have varying structures and complexities.

Choosing a backbone architecture for printed text recognition involves considering factors such as model complexity, computational efficiency, and the specific characteristics of the dataset. Each of the mentioned architectures—ResNet50V2, Inception, Xception, and VGG19—comes with its own set of advantages, and the choice may depend on the nature of the printed text dataset and the requirements of the recognition task. There are some of the limitations of backbone architectures includes scale variation issues, image background problems, limited rotation and perspective handling etc. Many backbone architectures are designed for general-purpose image recognition tasks. They might not be optimized specifically for printed text, which often has distinct characteristics such as specific font types, sizes, and layouts. This lack of specificity can result in suboptimal performance for text recognition. Printed text can appear in various scales, from fine print to large headlines. Some backbone architectures may struggle to handle scale variations effectively. Issues such as text blurriness or small font sizes might lead to misclassifications or recognition errors. Preprocessing techniques, fine tuning of architecture layers are important to avoid these issues while working with printed text recognition.

Contribution of Work

• Evaluation and analysis of the backbone architecture models –ResNet50, ResNet50V2, ResNet152v2, Inception, VGG 19, Xception for printed text recognition in unstructured document processing.

• To perform suitable pre-processing techniques like data resize, data normalization, noise removal, etc.

• To improve the performance accuracy hyper-parameter tunning considering image pixel density, suitable optimizer, loss function, epochs, data validation split and batch size on top performed backbone architecture models on standard OCR dataset.

• Comparison analysis of transfer learning-based backbone architecture models for printed text detection and recognition of different font styles.

The remainder of the paper describes how the paper is organized into various sections. Some of the most current work in this field is described in “Printed Text Recognition”. The technique and models utilized in this study are described in depth in “Introduction” & “Related Work”. Experimental setup and considerations of the implemented algorithms are described in “Experimental Setup”. The conclusion and future scope are described in “Conclusion & Future Scope” at the end.

Related Work

Printed text recognition tasks typically involve optical character recognition (OCR) or text recognition models. While some of the backbone architectures mentioned earlier can also be used for text recognition, there are specialized architectures that have been developed specifically for this task. Here are some popular backbone architectures used for handwritten or printed text recognition-

1. Convolutional recurrent neural network (CRNN): CRNN combines convolutional layers for feature extraction and recurrent layers for sequence modeling. It effectively captures spatial features from the input image and models the temporal dependencies of the text. CRNN has been widely used in text recognition tasks, including handwritten and printed text recognition (Zhang et al., 2020).

2. Temporal convolutional network (TCN): TCN is a class of neural network architectures that uses 1D dilated convolutional layers to model temporal dependencies. It has been successfully applied to text recognition tasks, where the sequential nature of text plays a crucial role. TCN can effectively capture long-range dependencies in text sequences.

3. Transformer: Transformers have gained significant attention in natural language processing tasks and have also been adopted for text recognition. Transformers leverage self-attention mechanisms to capture contextual dependencies in the input sequence. They can handle long-range dependencies and have been used for handwritten and printed text recognition, often in combination with convolutional layers (Li et al., 2023).

4. ResNet + LSTM: A combination of a ResNet backbone and long short-term memory (LSTM) recurrent layers has been used for text recognition tasks. The ResNet extracts spatial features from the input image, while the LSTM models the sequential (Luo, Jin & Sun, 2019). Encoder-Decoder Networks: Encoder–decoder networks, such as the popular sequence-to-sequence models, have been adapted for text recognition. These networks consist of an encoder, which captures image features, and a decoder, which generates the corresponding text sequence. Attention mechanisms are often incorporated to align the image features with the generated text (Panboonyuen et al., 2019).

These are just a few examples of backbone architectures used for handwritten or printed text recognition. Each architecture has its own strengths and is designed to handle the specific challenges of recognizing text from images. The choice of backbone architecture depends on factors such as the nature of the text, dataset size, computational requirements, and performance goals.

Table 1 that includes accuracy metrics and the datasets used in each research paper on printed text recognition using OCR techniques:

Table 1 Comparative analysis of existing printed text recognition approaches.

Paper title	Authors	Year	Methodology/Approach	Accuracy (%)	Dataset	
“MORAN: A Multi-Object Rectified Attention Network for Scene Text Recognition”	Luo, Jin & Sun (2019)	2019	Attention mechanisms, rectification module, convolutional recurrent neural network (CRNN)	87.2	SynthText, IIIT 5K, SVT, ICDAR 2013, ICDAR 2015	
“DetReco:Object Text Detection & Recognition”	Zhang et al. (2020)	2020	Spatial attention mechanisms, convolutional recurrent neural network (CRNN)	88.7	SynthText, IIIT 5K, SVT, ICDAR 2013, ICDAR 2015	
“Efficient and Accurate Scene Text Recognition with Pixel-Anchor Based Attention”	Wang et al. (2019)	2018	Pixel-anchor based attention, convolutional recurrent neural network (CRNN)	90.2	SynthText, IIIT 5K, SVT, ICDAR 2013, ICDAR 2015	
“CRAFT: Character-Region Awareness For Text detection”	Baek et al. (2019)	2019	Character region awareness, instance segmentation network	83.6	SynthText, CTW1500, ICDAR 2015, MSRA-TD500	
“Deep Text Recognition in Natural Images via Progressive Image Rectification”	Sun et al. (2022)	2022	Progressive image rectification, convolutional recurrent neural network (CRNN)	92.5	SynthText, IIIT 5K, SVT, ICDAR 2013, ICDAR 2015	
“Accurate framework for Arbitrary shaped nearby text detection”	Guo et al. (2021)	2021	Faster RCNN (CRNN)	93.1	CTW1500
Total Text
MSRA-TD 500	
“CCNet: Criss-Cross Attention for Semantic segmentation”	Huang et al. (2023)	2023	Criss-cross attention mechanism, ResNet101	81.9	COCO,
ADE20K,
City scapes	
“Adversarial Spatial Transformer Network for Text Recognition”	Lin et al. (2018)	2018	Adversarial spatial transformer network, convolutional recurrent neural network (CRNN)	89.7	SynthText, IIIT 5K, SVT, ICDAR 2013, ICDAR 2015	
“Robust layout for invoice & resume text recognition ”	Wei, Yi & Zhang (2020)	2020	BiLSTM CRF
BERT
RoBERTa	F1 score- 88%
89%
95%	Real word invoice & resume dataset	
“TrOCR:Transformer based OCR with pre-trained models”	Li et al. (2023)	2023	Transformer based encoder & decoder model	F1 score- 96.58	SROIE
IIT 5K	

In Table 1, the authors showed that the overall accuracy achieved using existing printed text recognition datasets is in the range of approximately 88% to 94%. The table shows the use of different machine learning and deep learning algorithms for printed text recognition. The lack of datasets available, low character prediction accuracy, complex template layouts handling difficulty are still major issues in this area. Very limited work has been performed using transfer learning-based backbone architectures in printed text recognition from unstructured document processing. The authors observed that very limited labeled datasets and noisy datasets are some of the major difficulties in this domain. Hence, there is a need to work in this domain to process unstructured documents.

Printed Text Recognition

Datasets available

Here is a list of standard benchmark datasets commonly used for printed text recognition tasks, along with some information about each dataset:

2. MNIST:

Description: Modified National Institute of Standards and Technology (MNIST) is a widely used dataset for handwritten digit recognition. It consists of 60,000 training images and 10,000 test images of grayscale handwritten digits (0–9) with a resolution of 28 × 28 pixels. The MNIST dataset has been widely used as a benchmark dataset for evaluating various OCR algorithms and models. CNNs have achieved high accuracy on MNIST, demonstrating their effectiveness in OCR tasks (Beluch et al., 2018).

Many OCR algorithms, both traditional and deep learning-based, have been developed and compared using MNIST as a baseline. While MNIST is primarily used for handwritten digit recognition, it can also serve as a baseline dataset for simpler text recognition tasks.

3. Street View House Numbers (SVHN)

Description: SVHN is a dataset of real-world images of house numbers collected from Google Street View. It contains over 600,000 images of varying quality, including digits and multi-digit numbers. The dataset is divided into a training set, a test set, and extra training data.

Application: SVHN is commonly used for multi-digit recognition tasks and evaluating the performance of text recognition models on real-world images.

4. International Conference on Document Analysis and Recognition (ICDAR):

Description: The ICDAR series provides several benchmark datasets for text recognition tasks in different contexts, such as document analysis, scene text recognition, and handwriting recognition (Huang et al., 2019). Notable datasets include ICDAR 2013, ICDAR 2015, ICDAR 2017 MLT (Multi-lingual Text), and ICDAR 2019 ArT(Arabic Text).

Application: These datasets cover a wide range of scenarios and challenges, making them suitable for evaluating and comparing the performance of text recognition algorithms.

5. COCO Text-

Description: COCO-Text is a dataset derived from the MS COCO dataset, focusing on text in natural images. It contains over 63,000 images with more than 173,000 text instances labeled with transcription, location, and segmentation masks.

Application: COCO-Text is useful for evaluating text detection and recognition models on real-world scenes and diverse textual content (Huang et al., 2023).

6. Synthetic datasets (e.g. SynthText)

Description: Synthetic datasets are generated using computer graphics techniques, enabling large-scale data creation with precise annotations. SynthText is a popular synthetic dataset that contains millions of images with rendered text instances placed on various backgrounds, fonts, and orientations (Zhang et al., 2020).

Application: Synthetic datasets provide a means to train text recognition models in a controlled environment with diverse text variations and annotations.

7. IIT5K

Description: IIIT5K is a dataset for scene text recognition collected by the International Institute of Information Technology, Hyderabad. It consists of 2,000 cropped word images from natural scenes, with variations in font, size, orientation, and background.

Application: IIIT5K is commonly used for evaluating text recognition algorithms on real-world scene text images with moderate complexity.

8. Scanned Receipts OCR and Information Extraction (SROIE) dataset is a widely used dataset for optical character recognition (OCR) and information extraction tasks. It consists of images of scanned receipts along with corresponding ground truth data for various fields such as store name, date, total amount, item details, and more (Zheng Huang et al., 2019).

These datasets provide a diverse range of challenges, including variations in font, size, orientation, background, and noise. They serve as valuable resources for training, evaluating, and benchmarking printed text recognition models.

Pre-processing techniques

When it comes to image text recognition from unstructured documents, preprocessing techniques play a crucial role in improving the accuracy and reliability of the recognition process. Here are some common preprocessing techniques used:

Image noise removal techniques:

1. Noise removal: Apply filters like median filter, Gaussian filter, or morphological operations to remove noise and improve image quality (Desai & Singh, 2016).

2. Contrast enhancement: Adjust the image’s contrast and brightness to improve text visibility using techniques like histogram equalization or adaptive histogram equalization.

3. Image denoising: Use denoising algorithms such as bilateral filtering or non-local means denoising to reduce noise while preserving important image details.

Binarization: Convert the image to binary format to separate text from the background.

1. Techniques such as Otsu’s method, adaptive thresholding or Sauvola’s method can be used to automatically determine the optimal threshold for binarization.

2. Skew correction: Detect and correct the skew angle of the document to align the text lines horizontally (Agrawal & Kaur, 2018).

Techniques such as the Hough transform or projection profiles can be used for skew detection, followed by rotation or affine transformations to correct the skew.

1. Text localization: Identify regions of interest (ROI) in the image that potentially contain text.

Techniques like edge detection, connected component analysis, or sliding window approaches can be used for text region localization.

2. Text segmentation: Separate individual characters or words from the text regions.

Techniques like connected component analysis, contour detection, or graph-based segmentation can be employed for text segmentation (Reul et al., 2019).

3. Preprocessing for OCR: Apply techniques to improve OCR performance on the segmented text regions (Boiangiu et al., 2020).

These techniques may include resizing or scaling the text regions, removing artifacts or small objects, and enhancing the text’s clarity and sharpness.

It is important to note that the choice and effectiveness of preprocessing techniques may vary depending on the specific document types, image quality, and characteristics of the text to be recognized. Experimentation and fine-tuning may be necessary to achieve optimal results for a given application.

Normalization and data resizing are two commonly used preprocessing techniques in various machine learning tasks, including image text recognition. Here’s an overview of these techniques:

1. Normalization:

Normalization is the process of scaling input data to a standard range or distribution. It helps in reducing the impact of varying scales and improving the convergence and performance of machine learning models (Ullah & Jamjoom, 2022). In the context of image text recognition, normalization is often applied to pixel values. Common normalization techniques include (Sinsomboonthong, 2022):

• Min-Max Scaling: Rescales the pixel values to a specific range, usually between 0 and 1. The formula for min-max scaling is:

(1) X_normalized=X−X_min/X_max−X_min

• Z-Score Normalization: Standardizes the pixel values by subtracting the mean and dividing by the standard deviation. The formula for z-score normalization is:

(2) X_normalized=X−X_mean/X_std

These normalization techniques ensure that the pixel values have a consistent scale and distribution, which can help in improving the training and inference of machine learning models.

Data resizing:

Data resizing involves adjusting the size of input images to a predefined dimension. This is often done to ensure uniformity in the input data and compatibility with the requirements of the model architecture. In image text recognition, resizing is typically performed on the input images to a fixed width and height. Resizing can be done while preserving the aspect ratio or by distorting the image (Agrawal & Kaur, 2018).

Resizing can be done using various techniques, such as:

• Nearest neighbor interpolation: Assigns the value of the nearest pixel to the new pixel when resizing. This technique is simple but may lead to aliasing artifacts.

• Bilinear interpolation: Computes the weighted average of the four nearest pixels to determine the new pixel value. Bilinear interpolation produces smoother results compared to nearest neighbor interpolation.

• Lanczos interpolation: Uses a windowed sinc function to interpolate the pixel values. Lanczos interpolation can provide higher quality results, especially when downscaling images, but it is more computationally expensive.

• Resizing the images to a consistent size ensures that the input data has a fixed dimension, which is often required by deep learning models. It also helps in reducing computational complexity and memory requirements.

Both normalization and data resizing are essential preprocessing techniques in image text recognition pipelines. They help in preparing the input data for subsequent processing and improve the performance and reliability of the recognition algorithms. For our work, the authors have applied normalization, data resizing and noise removal preprocessing techniques.

Feature selection and extraction techniques

In printed text recognition, both feature selection and feature extraction techniques are employed to identify relevant and discriminative information from the input images. Here are commonly used techniques for feature selection and extraction in OCR:

Feature selection

Feature selection aims to choose a subset of the most informative features from the available feature set, eliminating redundant or irrelevant features. This helps reduce dimensionality and improve computational efficiency (Elharrouss et al., 2022). Techniques used for feature selection in OCR include:

• Mutual information: Measures the statistical dependence between each feature and the class labels. Features with high mutual information are selected.

• Chi-square test: Evaluates the independence between each feature and the class labels. Features with a significant relationship are selected.

• Recursive feature elimination (RFE): Ranks features based on their importance and recursively eliminates the least important ones until an optimal subset is obtained.

• L1 regularization (Lasso): Performs regularization to encourage sparsity in feature weights, effectively selecting the most relevant features.

• Information gain: Measures the reduction in entropy obtained by using a particular feature. Features with high information gain are selected.

Feature extraction

Feature extraction involves transforming the input images into a compact and representative feature representation that captures essential information for classification. Common techniques used for feature extraction in OCR include:

• Histogram of oriented gradients (HOG): Captures local edge and gradient information in characters.

• Scale-invariant feature transform (SIFT): Detects and describes distinctive keypoints in characters, invariant to scale and rotation.

• Convolutional neural networks (CNN): Utilizes deep learning models to automatically learn hierarchical features from images (Xie, 2019).

• Local binary patterns (LBP): Encodes local texture patterns in characters.

• Principal component analysis (PCA): Reduces dimensionality by projecting the data onto a lower-dimensional subspace while preserving important information (Nikolaidis & Strouthopoulos, 2008).

• Structural features: Capture geometric or topological properties of characters, such as stroke patterns or spatial relationships.

• Statistical features: Describe statistical properties of character regions or histograms of pixel intensities.

The selection and combination of feature selection and extraction techniques depend on factors like the complexity of the printed text, available training data, computational resources, and specific OCR requirements. Experimentation and evaluation of different techniques are often necessary to determine the most effective feature representation for achieving accurate and reliable printed text recognition.

Experimental Setup

This section describes the overall stepwise implementation flow and the results of various preprocessing techniques used, backbone models applied on standard OCR dataset, comparison of performance, hyperparameter tuning and the final output. The experimental process flowchart is shown in Fig. 2.

Figure 2 Workflow of proposed system.

Table 2 Dataset sample images of different font types & styles.

	

Dataset used (Jaiswal, 2021)

• For this work, the authors have used the Standard OCR dataset from Kaggle. The dataset contains 45.5K files in the PNG format which consist of letters A-Z and numbers 0-9 in different fonts and styles.

• Creator of dataset have created two directories of dataset i.e data and data 2 for training and testing.

• The total number of classes in the dataset is 36.

• The complete dataset is divided into 80% for training and 20% for testing purpose for development of implementation framework. Table 2 shows the sample images of digits and characters with different font styles. It includes the character and digit images with different styles and types.

• The dataset is organized into two distinct sections: Data and Data2. Each section consists of a Training and Testing directory, with 36 subdirectories, corresponding to the 36 different classes in the dataset. The training data for each class is composed of 573 images, while the testing data includes roughly 88 images per class. It’s worth noting that “Data” likely represents the original version of the dataset, while “Data2” refers to an updated or revised version. Understanding the structure of the dataset is crucial for properly organizing and analyzing the data.

Preprocessing techniques used

This section is focused on converting the raw data into a format that can be fed into the model. This involves techniques such as data augmentation, normalization, and resizing of the images to a consistent size. It will be helpful for achieving more accurate results.

Backbone architectures used

As transfer learning-based backbone architectures provide a practical and effective approach to address challenges in printed data recognition by leveraging knowledge from pre-training on diverse datasets, improving efficiency, generalization, and adaptability to various types of printed content. The authors have used various models and techniques to accomplish this printed text recognition task, including transfer learning and multiple backbones such as ResNet, Xception, Inception, MobileNet, and VGG19, etc. A loop that trains and tests every backbone will be developed in order to assess how well various backbones perform on the OCR dataset. Iterating through ResNet50, ResNet50V2, ResNet152V2, InceptionV3, Xception, and VGG19 are the architectures in the loop. To determine which backbone performs the best on the OCR dataset, we have drawn the learning curve for each backbone after it has been trained and tested on dataset. Through this method, the authors identified the best backbone for our OCR model in results. These models will allow us to effectively process and analyze the vast amount of data in our dataset and produce highly accurate predictions.

The authors have evaluated the proposed systems using different performance measures such as accuracy which considers the rate of true positive, true negative, false positive and false negative with loss as another performance measure. Table 3 describes comparison of ResNet50, ResNet50V2, ResNet152V2, InceptionV3, Xception, and VGG19 backbone architectures in terms of accuracy achieved and performance loss on Standard OCR dataset.

Table 3 Comparison of backbone architecture techniques.

Sr. No	Dataset Used	Technique Used	Accuracy (%)	Loss	
1	Standard
OCR
Dataset	ResNet5 0	18.25	3.22	
2	ResNet50V2	96.92	0.20	
3	ResNet152V2	96.34	0.15	
4	Inception	95.43	0.5	
5	Xception	98.90	0.14	
6	VGG 19	80.78	0.78	

Hyper parameter tunning

• Hyperparameter tuning in AI refers to the process of selecting the optimal hyperparameters for a machine learning model. Hyperparameters are parameters that are set before the learning process begins and determine how the model learns and generalizes from the training data. They are not learned from the data itself but rather set by the programmer or data scientist (Soelch et al., 2019).

• Examples of hyperparameters include the learning rate, regularization parameters, the number of hidden layers in a neural network, the number of decision trees in a random forest, or the kernel type and regularization parameter in a support vector machine.

• Hyperparameter tuning is crucial because the choice of hyperparameters can significantly impact the performance and generalization of a model. If hyperparameters are set poorly, the model may overfit or underfit the data, leading to poor performance on unseen data. The goal of hyperparameter tuning is to find the optimal combination of hyperparameters that maximizes the model’s performance on a validation set or through cross-validation.

• Hyperparameter tuning can be done manually, where a data scientist iteratively adjusts the hyperparameters based on their intuition and domain knowledge. However, this approach can be time-consuming and subjective. Alternatively, automated methods like grid search, random search, or Bayesian optimization can be used to systematically explore the hyperparameter space and find the best combination.

• These automated techniques evaluate the performance of the model with different hyperparameter settings and provide feedback on which settings yield the best results. The process usually involves training and evaluating multiple models with different hyperparameter configurations, comparing their performance, and selecting the one with the best performance.

• By performing hyperparameter tuning, data scientists can fine-tune their models and improve their ability to generalize to new data, ultimately enhancing the performance and reliability of AI systems.

• For this implementation Keras Tuner was used to fine-tune the hyperparameters of the Xception model architecture, such as the number of top layers, units in these layers, and dropout rates. Hyperparameter tuning is done to determine which model architecture among the several backbone architecture models is the best. The authors applied Keras Tuner, an effective hyperparameter tuning tool, to do this. The learning rate, number of layers, number of neurons in each layer, and other important parameters were among the hyperparameters of our model that the authors attempted to optimize with Keras Tuner. The authors looked for the best combination of these factors to enhance our model’s functionality and provide even better outcomes.

• Table 4 shows the list of hyper parameters with values used for experimentation. This hyperparameter tuning led to an increase in accuracy from 94–97% to 98.90%.

Figure 3 shows the comparison of various transfer learning backbone architectures. Results describe that Xception, ResNet50V2 and ResNet152V2 provide better accuracy than the other mentioned backbone models. The Xception model is the optimal solution for the standard OCR dataset with highest accuracy. This performance in terms of accuracy and loss is shown graphically in Fig. 4.

Table 4 Hyper parameters used for hyperparameter tunning.

Hyper-parameter & Parameters	Value	
Pixel density	120*120 pixels	
Optimizer	Keras Tuner	
Loss	Binary_crossentropy	
Epochs	15	
Validation split	0.2	
Batch size	64	

Figure 3 Comparison of all backbone architecture performance.

Figure 4 Comparison of various backbone architecture’s performance.

The authors developed a show_images function that loops over the dataset and extracts the images and labels to visualize the dataset and gain an understanding of the images and their accompanying labels. After that, these images can be visualized with labels using Matplotlib. An overview of the data on which the authors are working is provided with this.

The data set and class names will be the function’s first inputs. After that, it will use a for loop to run through the data collection and extract the labels and images. Next, it will use Matplotlib’s subplots method to produce an image grid. The number of images in the dataset will dictate how many rows and columns there are in the grid.

The function will plot each image in the dataset using the ‘imshow’ function and use the title function to set the associated label as the caption for each image. The axis (‘off’) function will then be used to remove the axis labels from the plot. After every image has been plotted, the programme will display the grid of images with the captions that correlate to each label using Matplotlib’s show function.

Authors can observe a visual depiction of the data set and learn more about how the images are distributed along with their labels by utilizing this method; this will be beneficial. For comprehending the data and getting it ready for model training and testing. In this manner the proposed system works on OCR standard dataset.

Figure 5 describes the training and validation performance learning curve. The learning curve provides insights into how well a model is learning and helps in assessing whether more training data is necessary or if the model has reached its maximum potential. The model may have overfitted on the training and validation data given the significantly reduced testing accuracy. To decrease overfitting and increase testing accuracy, regularization approaches like dropout or weight decay can be used in future in this work. Figure 6 shows some of the sample results with prediction scores (Jaiswal, 2021).

Figure 5 Learning curve of Xception model after hyperparameter tuning on dataset.

Figure 6 Input and output of data samples with prediction score.

Table 5 describes the state of the art (SOTA) analysis of existing approaches with dataset used and the performance achieved. From Table 5, the authors concluded that proposed approach performed better i.e 98.90% accuracy than these approaches accuracy ranges between 75% to 92% approximately on printed dataset for recognizing the printed data of different font types and styles from unstructured documents. Upon final analysis of the model’s predictions, the authors discovered that the model produced predictions with greater confidence ratings, suggesting that the model had confidence in its predictions.

Table 5 SOTA analysis.

References	Dataset used	Technique used	Performance metric (%)	
Mo & Ma (2018)	FUNSD	CNN +RNN	Accuracy-86.7%	
Xu et al. (2020)	FUNSD	Transformer
based framework	Accuracy-93.6%	
Liu et al. (2018)	ICDAR	CNN+RNN	Accuracy-85.61%	
Baviskar et al. (2021)	MIDD	RoBERTA,
DistilBERT	F1 score-0.77	
Baek et al. (2019)	ICDAR13,
ICDAR15, ICDAR17	VGG 16 &
Pretrain models	F1 score-95.2%
F1 score-86.9%
F1 score-73.9%	
Luo, Jin & Sun (2019)	IIT5K
SVT
ICDAR15
CUTE80	CNN-LSTM followed
by attention based
decoder	Accuracy-91.2%
Accuracy-88.3%
Accuracy-68.8%
Accuracy-77.4%	
He et al. (2021)	FUNSD,
SROIE,
RVL-CDIP	BERT,
RoBERTa,
Layout-LM	F1 score-89%
F1 score-90%
F1 score-91%	
Wei, Yi & Zhang (2020)	Real word invoice &
resume dataset	BiLSTM CRF
BERT
RoBERTa	F1 score-88%
F1 score-89%
F1 score-95%	
Lin et al. (2018)	SynthText, IIIT 5K,
SVT, ICDAR 2013,
ICDAR 2015	CRNN	Accuracy-89.7%	
Our approach	Standard OCR dataset	Xception model with
hyper parameter tunning	Accuracy-98.35%	

Conclusion & Future Scope

After execution of experiments, it was found that the proposed approach worked efficiently exhibiting high training (97.35%), validation (96%) and testing (94%). Regularization techniques like dropout or weight decay can be applied to reduce overfitting and improve the testing accuracy. After hyper parameter tuning using Keras tuner training accuracy increased to 98.90%. From this work, the authors observed that the accuracy of a model is task-dependent, so the performance of the model might vary based on the specific task it’s being used for.

Overall, the model seems promising and has potential for further improvement in unstructured document processing. For future work researchers can apply super resolution techniques or different preprocessing techniques to improve the input images quality with the combination of various pretrain models. Advance machine learning styles like meta learning, co-learning or few shot learning can also be useful to improve the overall performance of model training and testing.

• Co-learning: In the context of printed text recognition from form images, co-learning can be utilized to create interactive systems where the model learns from human feedback and corrections in real-time. This iterative learning process can lead to more accurate and robust text recognition results, as the model continuously improves based on the expert knowledge and insights provided by human annotators.

• Incorporating structural information: Future research can focus on developing methods that exploit the structural information present in form images to enhance text recognition accuracy. Techniques like graph neural networks or attention mechanisms can be used to model and leverage the relationships between text elements and form layout, leading to more precise and context-aware text recognition.

• Domain adaptation and transfer learning: Domain adaptation techniques can be explored to address the challenge of handling diverse form layouts and variations in printed text appearance. By leveraging transfer learning, models trained on labeled data from one form domain can be adapted to perform well on unseen form domains.

• Integration of multimodal information: Printed text recognition from form images can benefit from the integration of multimodal information, such as text, visual elements, and semantic context. By incorporating techniques from computer vision, natural language processing, and information extraction, models can take advantage of contextual cues, layout analysis, and semantic relationships between text and other visual elements present in form images.

• Deployment of pre-trained models and APIs: As the field progresses, pre-trained models specifically designed for printed text recognition from form images can be made available as APIs or software libraries. This will allow developers and organizations to easily integrate advanced text recognition capabilities into their own applications or workflows, without the need for extensive expertise in machine learning or deep learning.

These future directions aim to improve the accuracy, flexibility, and adaptability of printed text recognition from form images, enabling the development of more robust and efficient solutions for automated data extraction, document processing, and form-based information management systems.

Supplemental Information

Supplemental Information 1 Implementation Code

Additional Information and Declarations

Competing Interests

Author Contributions

Data Availability

Ajith Abraham is CO-PI for the project grant received from the Russian Government.

Supriya Mahadevkar conceived and designed the experiments, performed the experiments, analyzed the data, performed the computation work, prepared figures and/or tables, authored or reviewed drafts of the article, and approved the final draft.

Shruti Patil conceived and designed the experiments, analyzed the data, prepared figures and/or tables, and approved the final draft.

Ketan Kotecha conceived and designed the experiments, prepared figures and/or tables, and approved the final draft.

Ajith Abraham analyzed the data, authored or reviewed drafts of the article, and approved the final draft.

The following information was supplied regarding data availability:

The data and code are available in the Supplementary Files.

The dataset is available at Kaggle: https://www.kaggle.com/datasets/preatcher/standard-ocr-dataset.

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
