# Peer review of "A comparison of deep transfer learning backbone architecture techniques for printed text detection of different font styles from unstructured documents"

_PeerJ Computer Science, doi:10.7717/peerj-cs.1769_

## Round 0.1 · original submission · Minor Revisions

The paper has some minor sentence corrections and equations to be cited properly. The reviewers have given comments that must be addressed well.

**Language Note:** The review process has identified that the English language must be improved. PeerJ can provide language editing services - please contact us at copyediting@peerj.com for pricing (be sure to provide your manuscript number and title). Alternatively, you should make your own arrangements to improve the language quality and provide details in your response letter. – PeerJ Staff

·

Basic reporting

A comparison of deep transfer learning backbone architecture techniques for printed text detection of different font styles from unstructured documents- This article contains desirable scientific contributions with readable language contents with neat presentations.

Experimental design

Motivation and execution of the article is clear.

Validity of the findings

Conclusions are well stated, linked to original research question & limited to supporting results.

Reviewer 2 ·

Basic reporting

1) A thorough language revision will elevate the overall quality of this valuable research work.
2) Figure 6 (Sample results with prediction score) is blurred. Prediction results are not readable.

Experimental design

The authors have mentioned various models and techniques employed for printed text recognition, which is commendable. However, a critical improvement would be to provide more comprehensive details about the specific architectures used, such as the layers, configurations, and modifications made to these models. This information is essential for readers to understand and replicate the methodology effectively. A more in-depth explanation of the architectural choices would enhance the paper's clarity and value.
Additionally, it's advisable to include a more insightful analysis of the methods' strengths and limitations.

Validity of the findings

1) The author's presentation of result graphs is a significant contribution to the research paper; however, their lack of detailed explanations is a notable shortcoming. To improve the paper, the author should provide comprehensive explanations for each graph, clarifying the key trends, patterns, and insights they reveal. This will help readers better understand the significance of the results and their implications for the study. Furthermore, the author should consider elaborating on the methods and data analysis techniques used to generate these graphs. This added context will enhance the overall clarity and credibility of the research findings and foster a deeper comprehension of the study's outcomes.

2) The inclusion of hyperparameter values for tuning in the paper is valuable; however, their lack of detailed explanations is a critical gap. To enhance the paper's clarity and utility, the author should provide thorough explanations for each hyperparameter, describing their roles and impacts on the model or algorithm. This will empower readers to comprehend the significance of these choices and their influence on the research outcomes. Additionally, elaborating on the reasoning behind the specific values selected for hyperparameters would further improve the paper's rigor and transparency.

Reviewer 3 ·

Basic reporting

1. The paper is organized well yet the content is not properly written in the sections. The manuscript contains numerous instances of redundancy and requires substantial rewriting.

2. In line 68, the author mentions a list of backbone architecture to be used in printed text recognition however there are no references for the same.

3. In the contribution of the work point 3 at line 133 is unclear. Hyperparameter tuning is the part of implementation that is generally adopted and hence it cannot be a contribution.

4. The paper reports the recognition rate for unconstrained text as it only recognizes characters with font variance. The word unconstrained text is used with respect to handwriting and document layout. Hence the author should reconsider the title of the manuscript.

5. The introduction does not specify anything about the state-of-the-art challenges in the domain of printed character recognition.

6. The objective of the work is not clear.

Experimental design

The figures are expressive in the manuscript but they are redundant. Figure 2 is just the elaboration of Figure 1.

In Table 1 there exist many methods to recognize printed text, and the proposed method is contributing in this regard. This objective is not clear.

In the Section Result and Discussion, there is no discussion about the failure cases and their justification.
In totality, this section needs to be rewritten. There should be comparisons in performance with the existing methods and discussions on the challenges associated with them.

Validity of the findings

1. The paper claims that there is no work reported in the domain of printed text recognition whereas there exist papers like :
Yingqiao Shi, Wenbing Fan, and Guodong Shi. 2011. The Research of Printed Character Recognition Based on Neural Network. In Proceedings of the 2011 Fourth International Symposium on Parallel Architectures, Algorithms and Programming (PAAP '11). IEEE Computer Society, USA, 119–122. https://doi.org/10.1109/PAAP.2011.23

Early-Modern Printed Character Recognition using Ensemble Learning (2017)

2. How this work is different from the work reported in the manuscript?

3. The current work does not present any novelty. It only shows an application of Deep Learning models on a predefined problem. Moreover, the paper lacks experiments on other mentioned datasets.

Additional comments

The work lacks objective and motivation.

---

## Round 0.2 · accepted · Accept

Readability is improved a good. All reviewers have good comments. Also, the results have improved a lot so I recommend an accept decision.

·

Basic reporting

A comparison of deep transfer learning backbone architecture techniques for printed text detection of different font styles from unstructured documents- Reviewer concerns are addressed meticulously. Readability is improved a good.

Experimental design

scope of the objectives and proposed methods are clearly mapped and desirable contributions present

Validity of the findings

No Comments

Reviewer 2 ·

Basic reporting

This article contains desirable scientific contributions with readable language contents with neat presentations.

Experimental design

Motivation and execution of the article is clear.

Validity of the findings

no comment

Reviewer 3 ·

Basic reporting

All the comments are implemented.

Experimental design

The graphs and the results are properly drafted.

Validity of the findings

All figures and necessary tables are updated.